# The Effect of Calcium-Silicate Cements on Reparative Dentinogenesis Following Direct Pulp Capping on Animal Models

**DOI:** 10.3390/molecules26092725

**Published:** 2021-05-06

**Authors:** Mihai Andrei, Raluca Paula Vacaru, Anca Coricovac, Radu Ilinca, Andreea Cristiana Didilescu, Ioana Demetrescu

**Affiliations:** 1Division of Embryology, Faculty of Dental Medicine, Carol Davila University of Medicine and Pharmacy, 8 Eroii Sanitari Boulevard, 050474 Bucharest, Romania; mihai.andrei@umfcd.ro (M.A.); raluca.vacaru@umfcd.ro (R.P.V.); anca.coricovac@umfcd.ro (A.C.); 2Division of Biophysics, Faculty of Dental Medicine, Carol Davila University of Medicine and Pharmacy, 8 Eroii Sanitari Boulevard, 050474 Bucharest, Romania; radu.ilinca@umfcd.ro; 3Department of General Chemistry, University Politehnica Bucharest, Spl. Independentei 313, 060042 Bucharest, Romania; I_demetrescu@chim.upb.ro; 4Academy of Romanian Scientists, 3 Ilfov, 050044 Bucharest, Romania

**Keywords:** biomaterials, direct pulp capping, dental pulp, dentin bridge, calcium-silicate cements

## Abstract

Dental pulp vitality is a desideratum for preserving the health and functionality of the tooth. In certain clinical situations that lead to pulp exposure, bioactive agents are used in direct pulp-capping procedures to stimulate the dentin-pulp complex and activate reparative dentinogenesis. Hydraulic calcium-silicate cements, derived from Portland cement, can induce the formation of a new dentin bridge at the interface between the biomaterial and the dental pulp. Odontoblasts are molecularly activated, and, if necessary, undifferentiated stem cells in the dental pulp can differentiate into odontoblasts. An extensive review of literature was conducted on MedLine/PubMed database to evaluate the histological outcomes of direct pulp capping with hydraulic calcium-silicate cements performed on animal models. Overall, irrespective of their physico-chemical properties and the molecular mechanisms involved in pulp healing, the effects of cements on tertiary dentin formation and pulp vitality preservation were positive. Histological examinations showed different degrees of dental pulp inflammatory response and complete/incomplete dentin bridge formation during the pulp healing process at different follow-up periods. Calcium silicate materials have the ability to induce reparative dentinogenesis when applied over exposed pulps, with different behaviors, as related to the animal model used, pulpal inflammatory responses, and quality of dentin bridges.

## 1. Introduction

Dental pulp vitality preservation is a necessity for prolonging a tooth’s life in the oral cavity. Dental pulp can be directly exposed in an oral environment as a result of deep carious lesions caused by cariogenic microorganisms, traumatic injuries or iatrogenic factors that can lead to bacterial infiltration, inflammation and infection [1,2]. Maintaining pulp vitality is necessary to avoid further complications that can lead to endodontic therapy or tooth extraction [3].

Pulp vitality can be maintained in certain situations by a stimulatory treatment that involves the activation of the dentin-pulp complex so as to produce reparative dentin at the level of the pulpal involvement area. In such situations, dental pulp protection against bacterial infiltration or toxicity of restorative dental materials, and dentin layer healing can be achieved by direct pulp capping (DPC) procedure, which involves the placement of a pulp-capping agent at the level of exposure [4].

Successful DPC consists in maintaining the vitality and functionality of the pulp and in new dentin bridge formation [5]. In addition to the size and depth of exposure, the presence of microorganisms at the site of exposure or the patient’s age [6], the type of biomaterial used for DPC plays an important role in the prognosis and success of the procedure [4,7].

### 1.1. Molecular Mechanisms in Dentinogenesis

The dentin–pulp complex is located inside the tooth. Both tissues, dentin and pulp, are embryologically developed from the dental papilla of the neural crest’s ectomesenchyme [8]. Dentin is a mineralized connective tissue, harder than the regular bone and less hard than the enamel, consisting mainly of calcium hydroxyapatite (Ca_10_(PO_4_)_6_(OH)_2_) and a collagenous matrix [9]. The pulp is a soft connective tissue that contains different cells such as odontoblasts, fibroblasts and undifferentiated mesenchymal cells [10], located in the pulp chamber and in the root canal, being isolated and protected from external factors by dentin, which surrounds it. Besides sensory, nutritional and defensive roles [11], dental pulp’s prime function is to secrete dentin due to the external odontoblastic cell layer [12]. 

During odontogenesis and up until the end of apexogenesis, primary dentin is formed, representing the majority of the circumpulpal dentine matrix [9]. Secondary dentin is formed in physiological conditions by continuous deposition of dentin after root development is completed, throughout life, as long as the tooth preserves its vitality [13]. Although slightly different, both primary and secondary dentin have regular, tubular structures comprising of intertubular and peritubular dentin. Tertiary dentin, a more dystrophic, sometimes atubular matrix, is formed specifically at the pulp-dentin interface following different pathological processes, either carious or traumatic. The intensity of the environmental stimuli may vary and produce two types of tertiary dentin. In response to a mild stimulus (e.g., slowly progressing caries), post-mitotic odontoblast cells are preserved and stimulated to secrete reactionary dentin [14]. If the stimulus is stronger (e.g., rapidly progressing deep caries, tooth cavity preparation or traumatic injury), the odontoblast cells are destroyed, and a much complex process is generated, which involves mitosis, chemotaxis, migration, adhesion and differentiation of mesenchymal stem cells to form a new generation of odontoblast-like cells, which will eventually secrete reparative dentin [15]. These cells may originate from the immune system, subodontoblastic cell layer, or from pulp fibroblasts, and their origin may influence the cell phenotype and further cellular interactions [14,16,17].

The differences regarding the morphology of the tubular structure of the three types of dentine are thought to be due to molecular substrate [15]. In fact, dentin is a reservoir of bioactive molecules, such as growth factors, neurotrophic factors, neuropeptides and cytokines, sequestered in extracellular matrix during dentinogenesis and that can be solubilized in pathological conditions: carious, traumatic or iatrogenic [18]. These bioactive molecules are signaling molecules that lead to receptor phosphorylation and modulate several signaling transduction pathways after binding, essential for dentinogenesis being MAPK (mitogen-activated protein kinase) and PI3K/AKT/mTOR (phosphatidylinositol 3-kinase/protein kinase B/mechanistic target of rapamycin), pathways that are engaged in cell proliferation, adhesion, migration and apoptosis. Among the MAPK family, p38 MAPK has the pivotal role of activating odontoblast’s secretory activity and is involved in repair and regeneration processes [18,19].

Vital pulp therapies and pulp-capping materials enhance biological responses and favor tertiary dentinogenesis by stimulating odontoblast’s secretory activity; therefore, further research is needed to understand specific molecular mechanisms and how to use them therapeutically. 

### 1.2. Calcium-Silicate Biomaterials as Pulp Capping Agents

A variety of biomaterials have been proposed and developed over time for pulp-capping procedures. Pulp-capping materials must ensure pulp regeneration and induce the formation of a hard tissue barrier following odontoblasts’ activation. Materials such as calcium hydroxide (CH) [20,21], zinc oxide eugenol cement [22], resin-modified glass-ionomer cement [23], MTA [24,25], adhesive systems [26], enamel matrix derivative [27], collagen [28], formocresol [29] and hydroxyapatite [30] have been proposed as pulp-capping agents over time.

In addition to the nature of the pulpal insult, the prognosis of success in pulp capping is greatly influenced by the type of pulp-capping agent used. An ideal pulp-capping material has to present biocompatibility, good adhesion to the dental hard tissues, compatibility with the restoration materials it comes in contact with, good marginal sealing, insolubility in tissue fluids, easy handling and manipulating, short setting time, proper mechanical properties, radiopacity, antimicrobial activity and possibly low cost [31,32].

In the past, formocresol has been an agent of choice for pulp therapy, with very controversial results, due to its cytotoxic, genotoxic and carcinogenetic effects [33]. Zinc oxide eugenol cement has also been proposed as a pulp-capping agent, but its effectiveness is questionable, especially because of eugenol’s high cytotoxicity and interfacial leakage [34,35]. Although eugenol is considered to have antibacterial properties, which might have been considered as an advantage of this material, it has been observed that its availability diminishes dramatically within time [36]. Furthermore, in human studies, it showed chronic pulpal inflammation and lack of healing or dentin bridge formation [37], as opposed to CH.

CH has been the gold standard for several decades since its introduction in 1921 [38], but its shortcomings such as lack of adhesion to dentin or resin restorations, poor mechanical properties, bacterial infiltration, tunnel defects in dentin bridges and pulpal resorption [39,40,41] led to the choice of clinicians towards new, much more elaborate materials such as MTA and later, materials derived from MTA: calcium silicate, calcium phosphates or calcium aluminate-based materials.

These biomaterials are hydraulic calcium-silicate cements that are derived from the original PC, and their composition is detailed in Figure 1. PC and MTA have similar compositions, while the other materials are modified or hybrid materials for mechanical and biological improvements. Due to their bioactivity, they can provide a much better dentin bridge formation than CH and a lower degree of pulpal inflammation [42]. The setting reaction known as hydration is a chemical reaction that involves the major compounds in dissolution and precipitation processes, resulting in different hydrates. This reaction can occur in wet environments. The setting reaction implies the reaction of calcium silicate with water forming calcium-silicate hydrate and calcium hydroxide [43].

### 1.3. Portland Cement (PC)

PC is the most common cement of general use and a basic ingredient of concrete. PC is composed of lime, silica, alumina, ferric oxide and other compounds [44]. PC has the following major phases: dicalcium and tricalcium silicate (as major constituents), tricalcium aluminate and tetracalcium aluminoferrite [45]. Compared to similar conventional dental materials, PC does not contain radiopacifying agents. PC also contains arsenic, which raises major concerns regarding its use due to its possible toxic effects [46]. However, arsenic levels are low and vary depending on the type of PC [47,48]. With this respect, the amount of arsenic released is not a reason for its contraindication [49].

Given the low cost of PC and the fact that it has an almost identical composition to MTA [50] and similar antibacterial effects [51], it can be considered a more economical alternative. In various studies, ordinary PC has been modified with different compounds: bismuth oxide, iodoform or zirconium oxide for radiopacification [52]; calcium chloride, which decreased the setting time, maintained the pH at high values, decreased solubility and reduced the quantity of water necessary for mixing, therefore, lowering the material’s permeability [53]; CH for antibacterial activity [45]. However, these changes in the material’s composition can affect the physico-chemical properties and bioactivity of the cement.

### 1.4. Mineral Trioxide Aggregate (MTA)

MTA was introduced in 1993 [54]. Initially, MTA was introduced as an endodontic material for root filling treatment, as it offered the possibility of apexification in immature teeth for apical barrier formation or repairment of various defects, such as perforations or fenestrations [55,56]. The bioactivity of the material promotes stimulation and regeneration at the level of the dentin-pulp complex, making it suitable as a pulp-capping agent. Initial grey MTA was replaced by white MTA in 2004 for aesthetic considerations, as it lacked in tetracalcium aluminoferrite, therefore, having a lower iron oxide content [57]. MTA is presented as a fine powder that is mixed with water. Usually, the ingredients come in pre-dosed packages. In terms of micro- and macroscopic properties, MTA is almost identical to PC. MTA is mainly composed of Portland type I cement with the main phases of tricalcium and dicalcium silicate, tricalcium aluminate, tricalcium and silicate oxides, and bismuth oxide as a radiopacifying agent. Depending on the manufacturer, the setting reaction can vary from a long time (2.5 h in the case of ProRoot MTA) to a much shorter time (15 min in the case of MTA Angelus) [57]. Even if MTA remains the most representative hydraulic calcium-silicate cement in its class, drawbacks, such as prolonged setting time, which requires more treatment sessions, high cost and difficult handling [58], lead to the release of derivate materials. New modified materials based on calcium silicates have been formulated over time in response to the disadvantages of MTA, but also in order to improve the pulpal response.

### 1.5. TheraCal LC

One MTA derivative material designed for direct or indirect pulp capping is TheraCal LC (Bisco, Schaumburg, IL, USA), which is a light-cured resin-modified tricalcium-silicate material released in 2011. TheraCal LC contains mineral compounds (type III PC), radiopaque agent (barium zirconate), hydrophilic thickening agent (fumed silica) and resins (bis-phenyl glycidyl methacrylate and polyethylene glycol dimethacrylate) [59,60]. The setting reaction is based on light curing for 20 s for each layer of applied material. As hydraulic cement, it depends on the water up taken from the dentin moisture and its diffusion within the material. The advantages of this material compared to MTA are its short setting time, leading to fewer treatment sessions, easy maneuverability and handling. Being a resin-based material, which does not require any conditioning of the dentin surface, TheraCal LC can be bonded with various types of adhesives after application [61]. In vitro studies showed a very low cytompatibility of TheraCal LC [62] due to the fact that resin-based materials exhibited toxic effects on cultured odontoblast-like cells [63]. This risk was associated with the residual unpolymerized resin components that remained in contact with pulp tissue, monomers such as BisGMA, HEMA, TEGDMA, and UDMA being cited as having the lowest biocompatibility [64,65]. In the last five years, nanoparticles, such as carbonic nanomaterials, hydroxyapatite, silica, iron and titanium oxide, zirconia, have been used in dental materials, such as TheraCal LC, to enhance their physical, chemical, and biological properties [66].

### 1.6. Biodentine

Biodentine (Septodont, Saint-Maur-des-Fossés, France) is a tricalcium-silicate cement developed as a dentin substitute. Biodentine is a powder-liquid system. The powder has the following main phases: tricalcium silicate, dicalcium silicate, calcium carbonate and zirconium dioxide as radiopacifier [67]. The liquid consists of calcium chloride. After mixing the components, the setting time takes up to 12 min for the cement to harden [68]. Compared to CH, Biodentine has lower porosity, better mechanical strength and less solubility. The dentin bridges show no tunnel defects, having a better sealing ability. Biodentine can be used as a temporary enamel substitute for up to 6 months and, without any surface treatment, could be a permanent dentin substitute [69]. In the literature [70], Biodentine’s behavior is presented as a favorable repair material based on its biocompatibility assigned to hydroxyapatite crystals’ deposition on the surface, which is in direct contact with tissue fluids [71]. 

### 1.7. Bioceramic-Based Materials

Initially developed for various other purposes, such as joint replacements, blood vessel prostheses or heart valves, they had been promoted as efficient and biocompatible endodontic materials, with high success rates [72]. Endodontic bioceramics are not influenced by humidity or blood contamination [73,74], are stable and insoluble over time, providing an adequate seal [75]. Before setting, bioceramics manifest an antibacterial effect, and afterwards, they become biocompatible and bioactive [76]. Among these bioceramic endodontic materials, we can mention BioAggregate (Innovative Bioceramix, Vancouver, BC, Canada), with a composition similar to that of MTA’s. Its composition consists of tricalcium silicate, calcium phosphate and silicon dioxide and tantalum oxide as a radiopacifier [67]. BioAggregate consists of Ca, Si and O, with a lower content of Al and lack of heavy element contamination [77]. BioAggregate can promote adhesion, migration, attachment [78], odontoblasts’ differentiation [79] and mineralization by activating the MAPK pathway of the human dental pulp cells [80].

### 1.8. Premixed Materials

A series of premixed or ready-to-use materials have been developed; this category includes iRoot BP Plus (also known as EndoSequence root repair material/EERM; Innovative BioCeramix, Vancouver, BC, Canada), TotalFill BC RRM (FKG Dentaire SA, La Chaux-de-Fonds, Switzerland) or Endocem MTA (Maruchi, Wonju, Korea). Their advantage is that they have a uniform consistency, are not technique sensitive and lack waste [81]. Premixed bioceramics require moisture from the environment in order to harden.

iRoot BP Plus is a premix bioceramic thick/putty white paste material, radiopaque, composed of tricalcium silicate, zirconium oxide, tantalum pentoxide, dicalcium silicate, calcium sulfate, calcium phosphate monobasic and filler agents, aluminum-free, insoluble [67], which requires water to harden and does not shrink during setting, with excellent physical properties [82]. During the setting reaction, when it comes in contact with water and/or moisture, it produces CH [82]. iRoot BP plus is a biocompatible material with antibacterial properties capable of inducing new dentin bridge formation when applied as a DPC agent [83]. 

TotalFill BC RRM (root repair material) is a calcium-silicate-based material for root canal repair and filling material. TotalFill BC RRM is available in three consistencies: injectable paste, putty and malleable putty. According to the manufacturer, TotalFill BC RRM consists of tricalcium silicate, dicalcium silicate, zirconium oxide, tantalum pentoxide and anhydrous calcium sulfate. The material has a very good radiopacity and an adequate setting time [84]. However, the material showed less marginal adaptation and sealing ability when compared to ProRoot MTA [85].

Endocem MTA is an MTA-derived material for endodontic use. Endocem MTA’s composition consists of calcium oxide, aluminum oxide, silicate oxide, magnesium oxide and bismuth oxide [86]. Endocem MTA has a number of advantages such as a faster setting time than conventional MTA and a better washout resistance [87], less tooth discoloration than MTA [88], a low cytotoxicity effect for preosteoblastic cells [89], calcium release and the ability to form apatite-like structures [90].

These bio-inductive materials, based on calcium silicates, can be used in different endodontic procedures, depending on the nature of each material and its purpose: apexification and apexogenesis, root perforations repair, as direct and indirect pulp-capping agents, in pulpotomy or pulpectomy procedures, regenerative procedures and as root filling materials [67]. 

The purpose of this review is to highlight the effect of calcium-silicate cements in reparative dentinogenesis, providing a better insight on molecular mechanisms on the different types of calcium-silicate cements and on histological findings in animal models, following DPC capping with these pulp-capping agents.

## 2. Histological Findings in Animal Model Following Direct Pulp Capping Procedure

### Methodology

We performed an extensive search of the PubMed (Medline) database, limited from 1962 to October 2020, to identify studies that were eligible for the aim of this review. During the screening process, two reviewers (MA and RPV) extracted the data independently, using EndNote X8 and X9 (Clarivate Analytics) for reference management. The search strategy for PubMed used medical subject heading terms and their variations, as shown in Table 1. Case series and reports, letters to the editor, and reviews were excluded manually during the screening process. Reference lists of the included articles were hand-searched in order to identify further eligible studies. Afterwards, full-text articles were assessed for eligibility.

The studies were included if they met the following criteria:original scientific studies;studies performed on animal subjects;studies reporting DPC with hydraulic calcium-silicates cements;studies reporting histological assessments, with highlights on inflammatory cell infiltrate, pulp tissue disorganization, reparative dentin formation, quality of reparative dentin, defective area or cell inclusion, the impact of dentin fragments.

We considered a study ineligible for inclusion if any of the following criteria were met:case reports, case series, letters to the editor, reviews;studies performed in vitro;studies that included indirect pulp capping or pulpotomy on animal models;non-English language publications.

For each included study the following data were recorded: author, publication year, country, study design, animal species, sample size, follow-up time, intervention, materials used, outcomes assessed and results.

From the articles included for the review, only the part concerning the histological evaluations following DPC on animal models was selected. Additionally, each study had to involve the use of a pulp-capping agent that is a calcium-silicate-based material, regardless of whether the material was used as a tested material or as a control material. Studies reporting results of pulpotomies or indirect pulp capping were excluded. 

## 3. Results

A total of 26 publications were included in this review. The studies were performed on different experimental animals. We have included in our review 13 studies performed on rats or mice (Table 2), 9 studies on dogs (Table 3), 3 studies on minipigs (Table 4) and 1 study on primates (Table 5). The number of animals included in studies varied from 2 to 45, while the number of teeth on which DPC was applied varied from 19 to 128. All studies compared MTA with other pulp-capping agents, except for one study that compared MTA with MTA enriched with calcium chloride [92], one that did not include MTA in the materials tested [93] and one that did not have a comparison material for MTA [94]. In all selected articles, the pulp chamber was opened by mechanical exposure, using different shaped burs, files, probes or endodontic explorers. The cavities were prepared mainly on maxillary molars, except for one study that performed DPC on incisors [95], two on canines [96,97], one on anterior teeth [98], two on incisors, canines and premolars [92,99], two on incisors, canines, premolars and molars [100,101], one on incisors and molars [102], one on mandibular molars [103] and four did not provide this information [93,104,105,106]. The cavities performed on molars were primarily class I occlusal cavities, while for the other categories of teeth, class V cavities were prepared mainly on the buccal surface. The pulp exposure diameter varied from 0.4 mm to 1.5 mm, the majority being between 0.8 and 1 mm wide. Various restorative materials were used for cavity filling after the placement of the DPC agent, such as self-curing glass-ionomer cement, light-cured glass-ionomer cement, light-cured dental resin, flowable dental composite, zinc-eugenol oxide intermediate restorative material or silver amalgam. Follow-up periods of time varied among the included studies from a minimum of 7 days to up to 3 months. There were 11 studies [92,93,95,96,102,104,105,107,108,109,110] that had only one follow-up evaluation, 10 with two [97,98,100,101,106,111,112,113,114,115], 3 with three [99,103,116], 1 with four [117] and 1 with five [94] follow-up evaluations. After the animals were euthanized, the teeth were histologically processed so that they could be analyzed under light microscopy.

The histological evaluation of the dental pulp and of the new dental hard tissue formation, subsequent to the action of the DPC agent, was performed by the authors by scoring the following criteria: inflammatory cell infiltrate, pulp tissue disorganization, reparative dentin formation, quality of reparative dentin, defective area or cell inclusion, the impact of dentin fragments (dentin chips) into the pulp (Figure 2). Detailed results of the eligible studies are presented in Table 2, Table 3, Table 4 and Table 5, including histological evaluations, along with the type of DPC materials used, the animals and the teeth on which DPC was performed. 

## 4. Discussions

The main purpose of DPC is to maintain the full integrity of the pulpal tissue in different pathological exposure conditions. An ideal DPC material should not induce inflammatory pulpal reactions, which may lead to necrosis, and should provide a quality repair dentin at the level of the exposure site [118]. Calcium-silicate-based materials have good efficacy on dental pulp when used as DPC agents [119].

MTA’s novelty was represented by the setting reaction that could occur in an aqueous environment. Following the setting reaction, calcium oxide converts into CH and calcium ions are released, stimulating cell adhesion and proliferation and leading to an increase of the pH, which offers antibacterial activity [120]. MTA can induce and stimulate cells to form hard tissue deposition and promote mineralization [121]. MTA’s ability to release calcium ions can induce dental pulp stem proliferation [122]. MTA showed a relatively fast pulp response by deposition of newly formed hard tissue, with slight signs of pulp inflammation [94]. 

CH was considered a gold standard pulp-tissue-regenerating material due to its biocompatibility, highly alkaline pH, bactericidal effect and capability to induce tertiary dentin formation [123]. Furthermore, CH had high clinical success as DPC material, documented in studies that followed patients for more than 10 years [124]. However, due to its high alkalinity, it also induces necrosis and inflammatory pulp responses [38]. Other disadvantages are its high solubility [125] and its lack of adherence to hard tissues, failing to offer an optimal seal, although, by the time of its full dissolution, the dentin bridge seems to be fully formed [126]. There is also stated that the dentin bridge induced by CH presents “tunnel defects” [40]; however, there is data that shows that the aspect of these defects improved with the thickening of the dentin bridge [127]. Light-cured CH has a better success rate than conventional CH because it is less subject to the process of hydrolysis of CH in the presence of moisture [128]. MTA has a higher clinical success rate than CH [98] and can induce the formation of much thicker dentin bridges [105,129]. However, in a clinical study on human vital teeth, performing indirect pulp capping with MTA and CH, similar behavior of the two materials was observed after clinical and radiological assessment, for 24 months [130]. 

Conventionally, it was believed that CH irritates the pulp due to its alkaline pH and stimulated tertiary dentin deposition [131]. CH and MTA have similar mechanisms of action based on calcium oxide, which reacts to carbon dioxide in tissues. Calcite granulations are formed, and fibronectin accumulates to the site, favoring cellular migration, proliferation, adhesion and differentiation [132], which leads to hard tissue formation [133,134]. During this process, bioactive molecules are released, Bone Morphogenic Protein (BMP) and Transforming Growth Factor-Beta One (TBF-β1), which mediate pulp regeneration and are incorporated in the dentin matrix during dentinogenesis [131,135,136,137].

The studies we have included in this review showed that animals pulp capped with MTA showed more frequently complete formation of new dentin bridges, with tubular structures, superior morphology and a lower rate of pulpal inflammation and necrosis, when compared to CH [96,99,102,104,105,117]. Even though MTA is more expensive than CH [3], it may be more cost-effective due to its stable clinical results over time, compared to CH, which requires future costly reinterventions [138].

Some authors tested modified MTA cements by incorporating different additives in order to improve their physical and biological performances; however, the results we have collected do not entirely support this supposition, especially concerning histological results. The addition of calcium chloride or amorphous calcium lactate gluconate-based liquid accelerated the setting reaction and improved handling, while the addition of propylene glycol provided a higher pH, with antibacterial effect [139], better flowability and increased calcium release [140]. There were no significant differences concerning acute inflammation between MTA with or without calcium chloride as DPC agents; however, MTA with calcium chloride showed a higher chronic inflammatory response, and the quality of the dentin bridge was inferior [92]. MTA modified with melatonin, a highly lipophilic hormone, which acts as a circadian rhythm regulator and anti-inflammatory agent [141], showed no significant differences regarding hard tissue deposition and pulp inflammatory response when compared to conventional MTA [107].

PC, with a similar composition as MTA [142,143], can be modified to achieve better performance through the addition of 20% bismuth oxide for radiopacity [106], the addition of antibacterial agents such as 2% chlorhexidine solution [93], calcium chloride to accelerate the setting reaction by increasing the hydration reaction and methylcellulose to avoid early washout following its application to the dentin [144]. PC cement is biocompatible and favors cellular attachment and growth [145]. PC can be considered as a low-cost substitute to MTA [146]. When PC with 20% bismuth oxide addition was compared to MTA and Port Cal (a material obtained from PC with 20% bismuth oxide and 10% CH powder added in the mixture), MTA had the best histological outcome followed by Port Cal, while PC had the highest inflammatory cell count. Although similar in composition, MTA’s fine and homogeneous particles’ content, as opposed to PC, offer this material its better neodentinogenetic characteristics [147], while the addition of CH in Port Cal may be the reason for less hard dentinal tissue deposition, due to its weaker and porous composition [106,147]. 

TheraCal LC, a hybrid DPC agent, induced dentinal hard tissue deposition at the level of exposure site with low inflammatory cell response [93,100]. This is due to the formation of CH following the hydration reaction, which created the premise of a high pH with antibacterial activity and the formation of calcium ions that induced morpho-differentiation and proliferation of odontoblast-like cells at the level of pulp exposure site [148]. Even though TheraCal LC contains resin in its composition, and there is a risk of remaining unpolymerized resin monomers, it has good biocompatibility compared to CH [149]. However, one study indicated that after 72 h, TheraCal LC showed a decrease in the percentage of cell viability, similar to CH [150]. Another study reported a lack of dentin bridge formation, probably due to pulp necrosis [93], and also the presence of necrotic pulp tissue and blood cloth underneath the exposure site at 70 days follow-up [100].

Biodentine and MTA had a similar effect concerning new dentin bridge formation capacity and induced fewer inflammatory signs when applied on human dental pulps following iatrogenic pulp exposure [151]. These aspects may be due to the fact that Biodentine induces odontoblastic differentiation from human dental pulp stem cells [152] and shows better biocompatibility and bioactivity than MTA and TheraCal LC [153]. The results related to Biodentine revealed that this material could induce thicker dentinal bridges than MTA and nHap, a nano-hydroxyapatite material [98], and a lower inflammatory pulp response [101]. 

iRoot BP Plus showed a slightly more favorable result as all specimens generated new dentinal tissue when compared to MTA [154]. In terms of bio-efficiency, both iRoot BP Plus and MTA had a similar effect on the dental pulps, leading to the deposition of a dentin structure with regular or irregular dentinal tubes pattern [155]. This aspect is mainly due to the ability of iRoot BP Plus to form apatite structures [82].

Bioactive glass materials [117] induced the formation of new dentin bridge formation one month after application, with low signs of pulpal inflammation and hard tissue deposition [112]. New endodontic cement (NEC) that contains several calcium compounds showed better results than MTA but without major differences. NEC specimens had a better organized odontoblastic layer with thick dentin bridges and a lack of pulp inflammation signs [96]. Nano-hydroxyapatite-based materials [98], Endocem, a fast-setting calcium-silicate-based cement containing zirconium oxide [108,110], ß-tricalcium phosphate [78], α-tricalcium phosphate-based materials [109] or Bio-MA [115,116] showed similar favorable pulp responses when compared to MTA.

Different materials such as dentin adhesives [99], light-cured resins [114], light-cured glass-ionomer cement [109], odontoblastic differentiating material [113] or biomimetic carbonated apatite [97] have been tested in an attempt to come up with new pulp-capping agents. In most cases, when compared to MTA, these materials showed less favorable results in terms of pulp inflammatory response and hard tissue deposition. New dentin bridge formation occurred in every specimen of the MTA group, while in the group with polymeric-based materials, there was no hard tissue deposition, even if the pulp tissue lacked any signs of inflammation [114] or, as shown in other studies, adhesive materials only induced formation in a few samples, and in several, there was inflammation and pulp necrosis [112]. Light-cured resin-based materials, such as restorative materials and dentin bonding agents, can induce pulpal adverse reactions due to the remaining incompletely polymerized monomer that may diffuse into the dental pulp and induce cell death [156] and to the shrinkage following the polymerization reaction that favors micro-infiltration [145]. Controversially, DPC with dentin adhesives on children’s permanent teeth showed a beneficial effect without signs of pulpal inflammation or necrosis and with a low failure rate [157], and in teeth pulp capped with a glutaraldehyde-based dentin-adhesive system, the results were favorable over a period of 6 months [158].

Summarizing, our results show that immediately after DPC, there is a substantial acute inflammatory response [103] underneath the pulp exposure, that tends to intensify over the next period of time and starts to diminish gradually afterwards [102,112], so as to become absent after one month following the intervention [96,97,100,101,110]. The lowest inflammation response was present in MTA samples and bioceramic materials, while CH samples showed a moderate response, and resin-based materials showed more persistent responses. The inflammatory process is beneficial and necessary for pulp healing and dental bridge deposition, as long as it is limited and does not lead to extensive necrosis and cell apoptosis [159]. Partial necrosis beneath the capping site was observed since the 1st day of follow-up [94], and the area extended in the next period [98]. Superficial necrosis was associated with the destruction of the odontoblastic cell layer [106] at 3 weeks after the intervention. At approximately 3 to 4 weeks, the odontoblast-like cells were present and organized in a palisade pattern underneath the injury site [110,116]. Additionally, disorganization of cell morphology adjacent to exposure site was observed at 3 weeks follow-up [103], but they became better organized within the following month [96]. Lack of necrosis was reported from the 30th day [101,107] and was associated with a continuous odontoblastic layer at 3 months follow-up [106]. Irrespective of the pulp-capping agent, during the first days following DPC, there were no signs of mineralized tissue at the exposure site. Only after 7 days, a newly formed mineralized matrix was observed, especially in MTA and Biodentine samples [103,112,115]. A dentinal tubular structure showed up during the 2nd week of follow-up [94], and by the 4th week, a heavier hard tissue deposition was observed, with both regular and irregular tubular patterns [93,100,112,114], that connected to the primary dentin [111]. Complete hard tissue bridge was reported in few studies, only in those that had longer follow-up periods [95,102]. The use of most resin-based materials, glass-ionomers and BCAp was not associated with signs of dental bridge formation [97,108,109,110], while CH had fewer spectacular results, as compared to MTA or bioceramic materials [104,105,106].

A possible limitation of our study is that we performed the search only in PubMed (Medline) database, and we excluded articles published in other languages than English. The results were heterogeneous, and a methodological inconsistency was observed throughout the included publications, concerning the animal model used, the pulp-capping agents, the follow-up periods and the outcomes assessed. Additionally, the evaluation criteria also made it difficult to perform a more concise analysis of the extracted data.

## 5. Conclusions

From in vivo experiments on animal models, it can be clearly seen that MTA remains the most commonly used and optimum DPC agent for vital pulp therapy, either as a material in the test group or as a reference material. MTA has taken over the gold standard from CH cements, remaining today a reference standard in this group of pulp-capping materials due to its properties in terms of biocompatibility, stimulation of odontoblasts, differentiation and proliferation of dental pulp cells in order to produce tertiary dentin, as shown on molecular and histological levels. As far as histological findings revealed, resin-based materials seem to be less suitable materials for pulp capping. More recently developed materials, such as bioceramic materials derived from MTA, show similar results and even surpass the ones already grounded for MTA. However, due to the diversity of animal models/teeth, type of pulp exposure, and DPC conditions, the translational success rate for some of the calcium-silicate cements might cover a wide range in human permanent/temporary teeth.

## Figures and Tables

**Figure 1 molecules-26-02725-f001:**
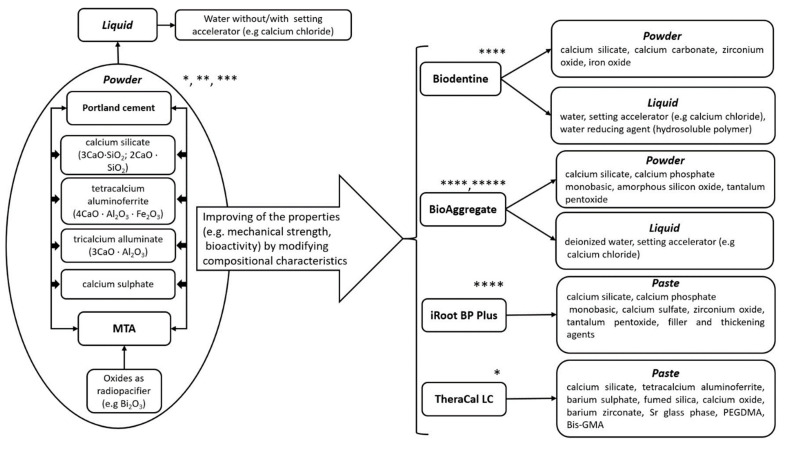
Composition of different calcium-silicate-based materials used as DPC agents. (* [60], ** [45], *** [50], **** [67], ***** [91]).

**Figure 2 molecules-26-02725-f002:**
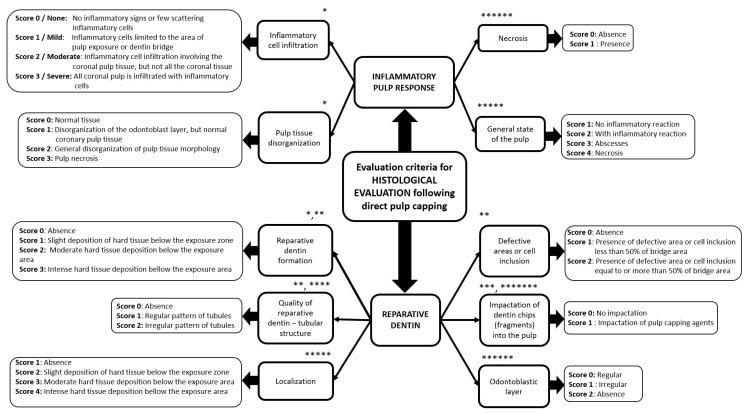
The histological evaluation criteria used for dental pulp characterization following inflammatory pulp response and new hard tissue formation, subsequent to the action of the DPC agent (* [103], ** [116], *** [115], **** [112], ***** [97], ****** [107], ******* [95]).

**Table 1 molecules-26-02725-t001:** The search strategy for PubMed used medical subject heading terms and their variations.

	Search Strategy Pubmed Database
**#1**	((((“Animals, Laboratory”[Mesh]) OR “Animals, Laboratory/drug effects”[Mesh]) OR (animal AND testing)) OR (laboratory animal)) OR (animal AND laboratory)
**#2**	(((“Dental Pulp Capping”[Mesh]) OR (Dental Pulp Capping)) OR (“Dental Pulp Capping/adverse effects”[Mesh])) OR ((((“Dental Pulp Exposure”[Mesh]) OR “Dental Pulp Exposure/drug therapy”[Mesh]) OR “Dental Pulp Exposure/physiopathology”[Mesh]) OR “Dental Pulp Exposure/adverse effects”[Mesh]) OR (dental pulp)
**#3**	(((((“Dentinogenesis”[Mesh]) OR “Dentinogenesis/drug effects”[Mesh]) OR “Dentinogenesis/physiology”[Mesh]) OR (dentinogenesis)) OR (tertiary dentin)) OR (dentin)
**#4**	(((((“Histological Techniques/analysis”[Mesh] OR “Histological Techniques/diagnosis”[Mesh] OR “Histological Techniques/drug effects”[Mesh])) OR (histologic)) OR (histology)) OR (histocytologic))
**#5**	((((((((pulp-capping agent[MeSH Terms]) OR ((((“mineral trioxide aggregate” [Supplementary Concept]) OR “Calcium Compounds”[Mesh]) OR “Calcium Compounds/adverse effects”[Mesh]) OR “Calcium Compounds/therapeutic use”[Mesh])) OR (MTA cement)) OR (Aggregate ProRoot)) OR (Tricalcium Silicate)) OR (Biodentine)) OR (“accelerated Portland cement” [Supplementary Concept])) OR (Portland cement)) OR (MTA)
**#6**	#1 AND #2 AND (#3 OR #4) AND #5 AND (English(Filter))

**Table 2 molecules-26-02725-t002:** Histological evaluation following DPC with calcium-silicate-based materials on rats/mice animal models.

Publication.	DPC Agent	Animal Type	Teeth Type	Histological Evaluation
Guerrero-Gironés, et al., 2020 [107]	Melatonin (5 mg, Sigma-Aldrich, St. Louis, MA, USA) MTA (ProRoot MTA, Dentsply Maillefer, Ballagues, Switzerland) MTA and melatonin Melatonin + melatonin	Sprague Dawley rats	First and secondary maxillary molars	30 days follow-up - All four groups of *MTA*, *melatonin*, *MTA and melatonin* and *melatonin + melatonin* showed vital pulps with a regular odontoblastic layer, lack of necrosis and new dentin bridge formation. - Melatonin’s dentinogenetic effect was no significantly different from that of MTA.
Paula, et al., 2020 [103]	White ProRoot MTA (Dentsply Tulsa Dental Specialties, Tulsa, OK, USA) Biodentine (Septodont, France) Positive control group: Glass ionomer cement (Ketac Fil Plus Aplicab, 3M Espe, USA) Negative control groups (No intervention performed)	Wistar Hun rats	First mandibular molars	3 days follow-up - Substantial amount of inflammatory cell infiltration was present in all groups, with complete pulp tissue disorganization, loss of connective tissue density and increase of calcium deposition in the *MTA* specimens. - *Biodentine* specimens were characterized by the presence of mild inflammatory infiltrate. 7 days follow-up - Matrix calcification was present in the *Biodentine* group with intense inflammatory infiltrate and increased cell morphology disorganization. - Slight inflammatory infiltrate and mineralized deposits with the maintenance of the tissue morphology were present in the *MTA* samples. 21 days follow-up - Pathological calcification and pulp tissue inflammatory cell infiltration, with dentin bridge formation and increased disorganization of cell morphology of the odontoblasts adjacent to the exposure site were observed in *MTA* samples. - *Biodentine* samples presented a normal structure under the dentin bridge, with the maintenance of the monolayer of odontoblasts in the pulp periphery, except for the exposure site.
Hanada, et al., 2019 [117]	Bioactive glass cement * (Nishika Canal Sealer BG/NCS-BG; Nippon Shika Yakunin, Yamaguchi, Japan), NSY-222-S—modified from NCS-BG, WMTA (WMTA ProRoot Dentsply, Tulsa Dental, OK, USA) CH (Dycal, Dentsply Caulk Milford, DE)	Wistar Rats	Maxillary first molars	1st day follow-up - A mild inflammation and no dentin bridge formation were observed in all groups 4 days follow up - Signs of slight pulp tissue inflammation were present and of a necrotic layer covering the entire exposed pulp and no signs of new dentin deposition. 7 days follow-up - A thin reparative dentin layer was present in all groups, and odontoblast-like cells were distributed with mild inflammation. 14 days follow-up - A thick reparative dentin with dentinal tubes was present, with mild inflammation and a decreasing tendency of the necrotic layer. - All tested materials had a similar response in new hard tissue deposition.
Trongkij, et al., 2019 [116]	White ProRoot MTA (Dentsply Tulsa Dental Specialties, Tulsa, OK, USA) Bio-MA (M-Dent/SCG, Bangkok, Thailand) Positive control group (uncapped pulp exposure) Negative control groups (intact teeth)	Wistar rats	Maxillary first molars	First-day follow-up - Regarding the inflammatory response, the two experimental materials had similar behaviors, presenting mild to moderate pulp inflammation with local disruption of the odontoblastic layer. - A mild to moderate inflammatory response in the *positive control* group. - No deposition of reparative dentin was present in any groups. 7 days follow-up - Moderate to severe inflammatory pulpal response was present in the *positive control* group. - Reduced inflammation in most specimens from both experimental groups with moderate hard tissue deposition, as well as a newly formed mineralized matrix. 30 days follow-up - Severe inflammatory response was present in the *positive control* group with moderate hard tissue deposition with non-tubular structure. - The presence of odontoblasts-like cells could be noticed under the newly deposited hard tissue layer in the *Bio-MA* and *MTA* groups. - Both *MTA* and *Bio-MA* induced hard tissue deposition, completely covering the exposed areas, with more than 50% tubular structure and cell inclusion, with no major differences between them in terms of quantity and quality of reparative dentin deposition. * All *negative control groups* displayed no inflammatory signs and intact odontoblastic layer at all follow-up times.
Trongkij, et al., 2018 [115]	White ProRoot MTA (Dentsply Tulsa Dental Specialties, Tulsa, OK, USA) Bio-MA ** (M-Dent/SCG, Bangkok, Thailand) Positive control group (pulp exposure without capping material) Negative control group (intact teeth with no preparation)	Wistar rats	Maxillary first molars	First-day follow-up - Mild to moderate inflammatory signs in the positive, *MTA* and *Bio-MA* groups. - Dental hard tissue deposition was absent in all groups. - Local disruption of the odontoblastic layer in *MTA*, *Bio-MA* and *positive control* groups. - Intact odontoblastic layer and lack of inflammatory signs in the *negative control* group. 7 days follow-up - The presence of a newly formed mineralized matrix was observed, and deposition of reparative dentin was present in some specimens from *MTA* and *Bio-MA* groups. - Only one sample in both testing groups displayed a continuous dentin bridge. - Diffused calcification below the exposure site in the *positive control* group. - Intact odontoblastic layer and lack of inflammatory signs in the *negative control* group.
Long, et al., 2017 [112]	MTA (ProRoot MTA, Dentsply, Sirona, Tulsa, OK) Novel bioactive glass: *** phosphate buffer solution solely (BG-PB) and phosphate buffer solution in addition with 1 wt.% sodium alginate (BG-PB-SA) Negative control group No DPC was applied, the cavity was sealed with a glass-ionomer cement (Fuji IX, GC International, Tokyo, Japan)	Wistar rats	Maxillary first molars	One week - Low inflammatory cell response was present in all experimental groups. - The *BG-PB-PA* and *MTA* groups showed a slight layer of newly generated matrix, while a mild hard tissue formation was observed in the *BG-PB* group. - Necrosis was present in the control group, with no hard tissue deposition. 4 weeks - No inflammatory response was observed in the majority of specimens, and only a few mild inflammatory responses occurred in the testing groups. - All testing groups showed heavy hard tissue deposition with regular tubular patterns in the newly formed dentin bridge, except for two *MTA* samples, with well-organized tubular dentin bridges. - An incomplete dentin bridge was present in the control group.
Liu, et al., 2015 [111]	iRoot BP Plus Innovative BioCeramix, Vancouver, BC, Canada) MTA (Dentsply, Tulsa Dental, Tulsa, OK, USA) Glass ionomer cement (*Control group*) (Fuji IX, GC International, Tokyo, Japan)	Wistar rats	Maxillary first molars	1 week follow-up - A similar inflammatory cell response was present in both *iRoot BP Plus* and *MTA* groups. - Regarding hard tissue deposition, all samples in the *iRoot BP Plus* group presented a mild hard tissue deposition, while the *MTA* group revealed a slight layer of newly generated matrix adjacent to the material in three-quarters of the specimens. - Mild to moderate inflammatory signs in the control group. 4 weeks follow-up - All *iRoot BP Plus* specimens and three-quarters of *MTA* specimens exhibited reparative dentin bridge formation, with a tubular dentin structure and the newly deposited dentin was connected to the primary dentin. - All specimens from the control group showed necrosis.
Kim, et al., 2015 [108]	Endocem Zr (Maruchi, Wonju, Korea) MTA (ProRoot, Dentsply, Tulsa Dental, Tulsa, OK, USA) Light-cured glass-ionomer cement (*Control group*) (Fuji II LC, GC, Tokyo, Japan)	Wistar rats	Maxillary first molars	4 weeks follow-up - Both *Endocem Zr* and *MTA* displayed reparative dentin with complete continuity underneath the pulp-capping materials at four weeks post-treatment, with no significant differences between the tested materials. - No hard-tissue presence was found in the control group, where teeth were DPC capped with a *light-cured glassionomer*
Lee, et al., 2014 [109]	α-tricalcium phosphate-based (α-TCP; Mediclus, Cheongju, Korea) MTA (ProRoot MTA, Dentsply, Tulsa Dental, Tulsa, OK, USA) Light-cured glass-ionomer cement (*control group*) (Fuji II LC; GC, Tokyo, Japan)	Wistar rats	Maxillary first molars	4 weeks follow-up - Tertiary dentin with complete continuity was formed underneath the pulp-capping agent in both testing groups. Odontoblasts-like cells were polarized and arranged in a palisade pattern. - No hard tissue deposition was found in the control group. - There were no significant differences between the two tested groups (*α-TCP* and *MTA*).
Moazzami, et al., 2014 [113]	Odontoblastic differentiating material (ODM) **** MTA (ProRoot MTA, Dentsply, Tulsa Dental, Tulsa, OK, USA) 2 control groups: teeth DPC with ODM without active ingredient and with a light-cured glass-ionomer cement (GC International, Tokyo, Japan)	Sprague Dawley rats	Maxillary molars	2 weeks follow-up - All specimens in the *MTA* group had vital pulps and, in some samples, mild inflammatory responses were present. - The *ODM* group presented a more intense inflammatory reaction, and only 80% of the pulps were vital. - Odontoblastic differentiation and reparative dentin formation were present in the *ODM* and *MTA* groups. - All specimens from both control groups were necrotic with no odontoblastic differentiation or hard tissue deposition. 2 months follow-up - Odontoblastic differentiation and reparative dentin deposition occurred in both groups with a maximum mean thickness of the dentin in the *ODM* group, where most of the pulps were vital, with partial necrosis beneath the capping site. - A well-organized tubular dentin bridge with predentin and the odontoblastic layer was present in *MTA* and *ODM* specimens on the floor of the pulp chambers.
Park, et al., 2014 [110]	Endocem (Maruchi, Wonju, Korea) MTA (ProRoot, Dentsply, Tulsa Dental, Tulsa, OK, USA) Light-cured glass-ionomer cement *Control group* (Fuji II LC, GC, Tokyo, Japan)	Rats	Maxillary first molars	4 weeks follow-up - The histological evaluation showed tertiary dentin formation with complete continuity beneath the pulp-capping agent in both testing groups, with no inflammatory or a mild inflammatory pulp tissue response. - Odontoblasts-like cells were present and arranged in a palisade pattern. - In the control group, there was no presence of tertiary dentin deposition.
Kuratate, et al., 2008 [94]	WMTA (white ProRoot MTA, Dentsply Tulsa Dental, Tulsa, OK) Negative control group	Rats	Maxillary first molars	1st day follow-up - A thin necrotic layer and a few inflammatory cells at the exposure site were present. 3 days follow-up - A slight to mild inflammation response was present. 5 days follow-up - New matrix formation was present at the exposure site. 7 days follow-up - A thin calcified bridge adjacent to the exposure site being observed in all samples. 14 days follow-up - Dentin bridge formation with a tubular structure was present in all samples with odontoblasts-like cells.
Simon, et al., 2008 [114]	MTA (ProRoot MTA, Dentsply, Tulsa Dental, Tulsa, OK, USA) Light-cured resin *Control group* (Point4, Kerr Hawe, Bioggio, Switzerland)	Mice	Maxillary first molars	2 weeks follow-up - A line with a high affinity for histological dye following the material contour was observed in the *MTA* group. 5 weeks follow-up - Samples from the *control group* showed normal pulp tissue with no inflammatory signs and lack of dentin bridge formation. - New dentin bridge formation was present in all specimens from the *MTA* group, and dentinal tubes with non-linear and interrupted trajectory could be observed in the matrix of the dentin bridges of three specimens.

* Bioactive glass cement [117]—Composition: NCS-BG: Paste A: fatty acid, bismuth subcarbonate, silicon dioxide; Paste B: magnesium oxide, purified water, calcium-silicate glass, silicon dioxide and others. NSY-222-S: Paste A: fatty acid, bismuth subcarbonate, silicon dioxide; Paste B: calcium oxide, purified water, calcium-silicate glass, silicon dioxide and others. ** Bio-MA [115]—Composition: calcium oxide, silicon dioxide, aluminum oxide, bismuth oxide, purified water and calcium chloride. *** Novel bioactive glass: BG-PB and BG-PB-SA [112]—Composition: powder: bioactive glass (82.36% SiO2, 15.36% CaO, and 2.28% P2O5); liquid: only phosphate buffer solution for BG-PB and phosphate buffer solution with the addition of 1 wt% sodium alginate for BG-PB-SA. **** ODM [113]—Composition: combination of active ingredients: 1, 25-dihydroxy vitamin D3, β-glycerophosphate disodium salt hydrate and dexamethasone; Polymer blend: sodium carboxymethylcellulose, hydroxypropyl methylcellulose and carbopol 934.

**Table 3 molecules-26-02725-t003:** Histological evaluation following DPC with calcium-silicate-based materials on dog animal model.

Publication.	DPC Agent	Animal Type	Teeth Type	Histological Evaluation
Zaen El-Din, et al., 2020 [98]	MTA (ProRoot White MTA, Dentsply, Sirona) Biodentine (Septodont, Saint-Maur-des-Fossés, France) Nano-hydroxyapatite (nHAP) * CH (Dycal, Dentply Sirona Endodontics)	Dogs	Anterior teeth	7 days follow-up - In the *CH, Biodentine* and *MTA groups,* there were mild signs of pulp inflammation, with partial pulp necrosis in some specimens and early signs of calcifications in one-third of samples. Some specimens showed partial tissue necrosis. - The *nHAP group* presented mild to none inflammatory pulpal tissue response, with partial tissue necrosis or early signs of mineralization in some samples. 3 months follow-up - Moderate inflammation was present in half samples of the *CH group*, while in the *MTA*, *Biodentine* and *nHAP groups,* pulp inflammation was absent or mild in two-thirds of the specimens, with tissue necrosis extended in some *MTA* and *Biodentine* samples. - Regarding hard tissue formation, in the *MTA* and *Biodentine* groups there were two-thirds of specimens with complete calcified dentin bridge, in the *CH group,* there were some samples with complete and more samples with interrupted calcified dentin bridge. - In the *nHAP group,* there were samples with both continuous and interrupted dentin bridges and a few samples with scattered calcific formations.
Akhavan, et al., 2017 [99]	MTA (ProRoot MTA, Dentsply, Sirona, Tulsa, OK), Dentin adhesives: *Clearfil S3/CS3, Bond* (Kuraray, Osaka, Japan); *Optibond* (FL/OBF, Kerr, Orange, CA, USA); *Single Bond/SB*, (3M, ESPE, MN, USA); *Clearfil* SE/CSE Bond (Kuraray, Tokyo, Japan) CH (Dycal, Dentsply, Germany)	Dogs	Premolars, canines, first, second and third incisors	7 days follow-up - *CSE* and *OBF* induced necrosis in one, respectively two specimens, while *SB* and *CS3* induced the hard tissue deposition in two specimens. - Inflammation was present in the *OBF* group in four samples, while SB induced stimulated odontoblastic generation. 21 days follow-up - Inflammation was present in 4 samples from the *OBF* group. - *SB* stimulated odontoblastic layer formation in 4 samples. 63 days follow-up - *MTA* specimens had the lowest inflammatory response, odontoblastic layer formation and the highest amount of hard tissue deposition.
Negm, et al., 2017 [106]	MTA (Endocem Maruchi, Korea) *Port Cal* ** PC with 20% bismuth oxide	Dogs	N/A	3 weeks follow-up - None of the three silicate-calcium-based materials presented new dentin bridge formation. - *Port Cal* specimens exhibited a continuous odontoblastic layer same as *PC with 20% bismuth* oxide group. - *MTA* samples showed destruction of the odontoblastic layer near the exposure site and some areas of superficial necrosis. 3 months follow-up - All groups had partial and complete new dentin bridge formation with the presence of a continuous odontoblastic layer. - *Port Cal* specimens displayed partial and complete dentin bridge formation, continuous odontoblastic layer and minimal inflammatory signs. - *The PC with 20% bismuth oxide* specimens showed the highest inflammation cell count, with complete or incomplete new dentin bridge formation. - *MTA* samples exhibited the highest scores for dentin bridge formation with the regularity of the pulp tissue architecture, with normal pulp and continuous odontoblastic layer.
Shi, et al., 2016 [95]	iRoot BP Plus (Innovative BioCeramix, Vancouver, BC, Canada) MTA (ProRoot MTA, Dentsply, Tulsa Dental, Tulsa, OK, USA)	Beagle dogs	Maxillary and mandibular incisors	3 months follow-up - Calcified bridge formation at the interface of pulp exposure could be observed in most of the specimens from both groups, with regular or irregular dentinal tubes pattern and dentin chips presence in some specimens. - One *MTA* specimen had connective tissue in the dentin bridge.
Danesh, et al., 2012 [97]	BCAp *** (biomimetic carbonated apatite) MTA (ProRoot MTA, Dentsply, Tulsa Dental, Tulsa, OK, USA)	Beagle dogs	Canines	7 days follow-up - *MTA* and *BCAp* groups had all pulps vital. - Some *MTA* samples with no complete dentin tissue formation. - *BCAp* samples with neither complete nor incomplete hard tissue bridges formation, with lateral deposition of hard tissue in 4 specimens. 70 days follow-up - All pulps from both groups were vital. - None of the specimens in the *BCAp* group presented new dentin bridge formation. - 6 specimens from the *MTA* group six had complete or incomplete hard tissue bridges, with mild or lack of pulp tissue inflammation. The bridges were thinly composed of dentin or irregular hard tissue.
Parirokh, et al., 2011 [92]	MTA (ProRoot MTA, Dentsply, Tulsa Dental, Tulsa, OK, USA) MTA With addition of calcium chloride	Dogs	Lateral incisor, canines and premolars	2 months follow-up - There were insignificant differences between the two groups, with no acute inflammation. - Specimens capped with *MTA with CaCl_2_* showed a higher chronic inflammatory response and less completely calcified dentin bridge formation and inferior quality.
Asgary, et al., 2008 [96]	MTA (ProRoot MTA, Dentsply, Tulsa Dental, Tulsa, OK, USA) CH (Dycal, LD Caulk, Milford, DE) NEC **** (Novel endodontic cement)	Beagle dogs	Canines	8 weeks follow-up - The *CH* group presented all pulps vital, pulp necrosis in two specimens and no complete calcified bridge formation. - All samples from both *MTA* and *NEC* groups had vital pulps with no sign of inflammatory response; complete dentin bridge formation was observed in 75% of the specimens. - The *NEC* group had a slightly better well-organized odontoblast-like cell layer and a sufficient thickness of the dentinal bridge.
Briso, et al., 2006 [104]	MTA (ProRoot MTA, Dentsply, Tulsa Dental, Tulsa, OK, USA) CH (Reagen, Quimibras, Rio de Janeiro, Brazil)	Mongrel dogs	N/A	60 days follow-up - In the *MTA* group, there were more specimens with complete and incomplete dental bridge formation than in the *CH* group. - There were fewer specimens in the *MTA* group with inflammatory cell response and necrosis. - The dentin bridge morphology was better in the *MTA* specimens with bridges composed of dentin associated or not to areas of irregular hard tissue deposition. - *MTA* specimens exhibited thicker hard tissue brides than *CH*.
Faraco and Holland, 2001 [105]	MTA (Dentsply Tulsa, Tulsa, OK, USA) CH (Dycal, L.D. Caulk, Milford, DE)	Dogs	N/A	2 months follow-up - *MTA* group had obviously better results than the *CH* group in terms of new hard tissue formation. - All *MTA* specimens exhibited hard tissue bridges with tubular dentin, while *CH* specimens presented a lower number of the newly formed structures. - The inflammatory pulp response was also better in the *MTA* group, while chronic inflammatory response and severe neutrophilic infiltrate were present in several cases in the *CH* group. - Absence of inflammatory infiltrate and microorganisms in the *MTA* samples.

* nHAP [98]—Composition: nano-hydroxyapatite crystals. ** Port Cal [106]—Composition: PC with addition of 10% calcium hydroxide and 20% bismuth oxide. *** BCAp [97]—Composition: white MTA with a sterile calcium- and magnesium-free phosphate-buffered saline solution for 40 days at 37 °C. **** NEC [96]—Composition: calcium hydroxide, calcium oxide, calcium phosphate, calcium sulfate, calcium silicate and calcium carbonate.

**Table 4 molecules-26-02725-t004:** Histological evaluation following DPC with calcium-silicate-based materials on pig animal model.

**Publication.**	**DPC Agent**	**Animal Type**	**Teeth Type**	**Histological Evaluation**
Li, et al., 2018 [100]	MTA (ProRoot MTA, Dentsply, Sirona, Tulsa, OK, USA), TheraCal LC (Bisco, Schaumburg, IL, USA) TCS 50 *	Gottingen minipigs	Incisors, canines, premolars and molars	7 days follow-up - Regarding the inflammatory response, all three materials showed similar behavior. - TCS50 presented a well-organized exposed pulp tissue, with no inflammatory signs in the deeper pulp area and a normal odontoblastic layer. - No hard tissue deposition was observed in any of the groups. 70 days follow-up - The specimens from all groups did not show inflammatory pulp reactions. - All three materials induced the formation of a complete mineralized tissue, with the highest thickness in the case of *TCS 50*. - In the *MTA group,* the matrix had a tubular structure with calcifications entrapped in the matrix. - In some samples from the *TheraCal LC* group, necrotic pulp tissue and a blood clot were present underneath the exposure site, but the deeper area of the reparative dentin presented continuous tubes.
Tziafa, et al., 2014 [101]	Biodentine *Experimental group* (Septodont, Saint-Maur-des-Fossés, France) MTA *Control group* (Angelus, Londrina, PR, Brazil)	Miniature swine pigs	Incisors, canines, premolars, molars	3 weeks follow-up - None of the two biomaterials showed a mature bridge formation. 8 weeks follow-up - Reactionary dentin formation around the exposure site associated with the newly formed matrix. - Both test groups presented a mineralized matrix formation in the form of a complete hard tissue bridge, with no detectable inflammatory responses or pulp necrosis.
Shayegan, et al., 2009 [102]	Beta-tricalcium phosphate (beta-TCP) (RTR, Septodont) White MTA (Dentsply, DeTrey GmbH) White PC (Cantillana, Belgium) CH (Dentsply, DeTrey GmbH)	Pigs	Incisors, maxillary and mandibular molars	3 weeks follow-up - *All beta-TCP* specimens presented new dentin bridge formation with a normal histological pulp pattern and odontoblastic layer. - *All WMTA* samples displayed complete thin calcified bridges with normal pulp tissue and lack of inflammatory response. - All specimens of the *white-PC* groups showed normal pulp architecture and complete calcified bridge formation. - In the *CH group,* one sample had moderate incomplete new dentin bridge formation, while the others presented complete calcified new dentin bridge formation. - There were no significant differences between the four groups in terms of inflammatory response and hard tissue deposition.

* TCS 50 [100]—Composition: Powder: 50 wt.% tricalcium silicate and 50 wt.% zirconium oxide; Liquid: calcium chloride.

**Table 5 molecules-26-02725-t005:** Histological evaluation following DPC with calcium-silicate-based materials on primate animal model.

Publication.	DPC Agent	Animal Type	Teeth Type	Histological Evaluation
Cannon, et al., 2014 [93]	TheraCal LC (Bisco, Schaumburg, IL, USA) Pure PC with 2% Chlorhexidine solution, Glass ionomer cement (Triage, Fuji VII, GC, USA) CH (Dycal, Dentsply)	Primate (Capucin Cebus Opella)	3 teeth in each quadrant	4 weeks follow-up - Only one sample in the *TheraCal LC* group lacked the deposition of hard tissue, most likely due to pulp necrosis. - A mixed result regarding the inflammatory response was present in all groups. - The newly formed dentin bridge had the highest average depth in the *TheraCal LC* group, followed by the *PC*, *glass-ionomer cement* and *CH* groups.

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
