# Peer review of "The Effect of Calcium-Silicate Cements on Reparative Dentinogenesis Following Direct Pulp Capping on Animal Models"

_molecules, 2021, doi:10.3390/molecules26092725_

Round 1

Reviewer 1 Report

This review described the effect of calcium-silicate cements on reparative dentinogenesis performed on the various animal teeth. It is a well-written paper and covers various in vivo studies. Also, the search strategy is appropriate. I do think it is worthy of being published in the journal. However, there are some problems which should be corrected.

Title
The manuscript deals only with animal studies. So, it is necessary to change the title more specifically.

Introduction
The aim of the review should be described at the end of the introduction section.

Line 51: Molecular pathways in dental pulp regeneration
In fact, it is about dentinogenesis or dentin-pulp complex regeneration, not pulp regeneration. So, the subtitle should be changed.

Line 100: Have zinc oxide eugenol cement and formocresol been proposed for the pulp capping materials? Please check if it is appropriate to describe.

Line 125: Indeed, PC is not recommended for pulp capping anymore because it contains toxic heavy metals such as arsenic. Therefore, it is required to describe the harmful ingredients in the PC.

Line 160: TheraCal LC has shown some cytotoxicity and inflammatory reaction in several previous studies. Therefore, it is necessary to indicate the shortcomings as well as the advantages.

Line 188:  iRoot BP Plus is a band name of the premixed type of materials. There are some other products in this category. So, I recommend changing the subtitle to other expressions such as premixed or ready-to-use material. Also, introduce other products as well as iRoot BP Plus. I suggest TotalFill RRM putty and Endocem MTA premix. Also, change the BioAggregate as well in the same way. Please refer to the following article. [Debelian et al. The use of premixed bioceramic materials in endodontics. Giornale Italiano di Endodonzia 2016:30:70-80]

Line 272: Table 2 is too long to read easily. I recommend dividing the table into 3 or 4 according to the animal model used. Also, list the studies according to the order of references.

The reference style does not meet the guideline of the journal. Please check all the references carefully again.

Author Response

This review described the effect of calcium-silicate cements on reparative dentinogenesis performed on the various animal teeth. It is a well-written paper and covers various in vivo studies. Also, the search strategy is appropriate. I do think it is worthy of being published in the journal. However, there are some problems which should be corrected.

Thank you! We sincerely appreciate your comments!

Title
The manuscript deals only with animal studies. So, it is necessary to change the title more specifically.

Thank you for your suggestion. Indeed, a more suitable title would include the fact that this is a paper about animal studies. The new title proposal is “The effect of calcium-silicate cements on reparative dentinogenesis following direct pulp capping on animal models”.

Introduction
The aim of the review should be described at the end of the introduction section.

We agree with this suggestion and we have moved the aim of the review at the end of the introduction section.

Line 51: Molecular pathways in dental pulp regeneration
In fact, it is about dentinogenesis or dentin-pulp complex regeneration, not pulp regeneration. So, the subtitle should be changed.

Following your suggestion, we modified the subtitle “Molecular pathways in dental pulp regeneration” to “Molecular mechanisms in dentinogenesis”.

Line 100: Have zinc oxide eugenol cement and formocresol been proposed for the pulp capping materials? Please check if it is appropriate to describe.

There are studies that mention zinc oxide eugenol and formocresol use as pulp capping materials, but due to their high cytotoxicity, they are not considered as suitable anymore. We took your advice into consideration and we added a paragraph about these 2 materials. Please find below the paragraph we have written:

In the past, formocresol has been an agent of choice for pulp therapy, with very controversial results, due to its cytotoxic, genotoxic and carcinogenetic effects [33]. Zinc oxide eugenol cement has also been proposed as a pulp capping agent, but its effectiveness is questionable, especially because of eugenol’s high cytotoxicity and interfacial leakage [34,35]. Although eugenol is considered to have antibacterial properties, which might have been considered as an advantage of this material, it has been observed that its availability diminishes dramatically within time [36]. Furthermore, in human studies it showed chronic pulpal inflammation and lack of healing or dentin bridge formation [37], as opposed to CH.

Line 125: Indeed, PC is not recommended for pulp capping anymore because it contains toxic heavy metals such as arsenic. Therefore, it is required to describe the harmful ingredients in the PC.

Regarding PC’s arsenic content, indeed that is a reason of concern for the dentists using this material and therefore we appreciate your suggestion and we have included a paragraph about this aspect. The paragraph is written below:

PC also contains arsenic, which raises major concerns regarding its use, due to its possible toxic effects [46]. However, arsenic levels are low and vary depending on the type of PC [47,48]. With this respect, the amount of arsenic released is not a reason for its contraindication [49].

Line 160: TheraCal LC has shown some cytotoxicity and inflammatory reaction in several previous studies. Therefore, it is necessary to indicate the shortcomings as well as the advantages.

Indeed, this is a good suggestion that we took into consideration. Please find below the paragraph we’ve added:

In vitro studies showed a very low cycompatibility of TheraCal LC [62], due to the fact that resin-based materials exhibited toxic effects on cultured odontoblast-like cells [63]. This risk was associated with the residual unpolymerized resin components that remained in contact with pulp tissue, monomers like BisGMA, HEMA, TEGDMA, and UDMA being cited as having the lowest biocompatibility [64,65].

Line 188:  iRoot BP Plus is a band name of the premixed type of materials. There are some other products in this category. So, I recommend changing the subtitle to other expressions such as premixed or ready-to-use material. Also, introduce other products as well as iRoot BP Plus. I suggest TotalFill RRM putty and Endocem MTA premix. Also, change the BioAggregate as well in the same way. Please refer to the following article. [Debelian et al. The use of premixed bioceramic materials in endodontics. Giornale Italiano di Endodonzia 2016:30:70-80]

Thank you for your suggestion. We have modified the subtitle to “Premixed materials” and also, we changed the subtitle referring to BioAggregate into “Bioceramic materials”.

Line 272: Table 2 is too long to read easily. I recommend dividing the table into 3 or 4 according to the animal model used. Also, list the studies according to the order of references.

We took into consideratiom your suggestion and divided Table 2 in 4 tables according to the animal model used.

The reference style does not meet the guideline of the journal. Please check all the references carefully again.

Thank you! We have checked the reference style and made the necessary modifications.

Reviewer 2 Report

The article is interesting and addresses an important subject in the field. However, it is necessary update references, mainly in introduction, since it is a literature review. 

In introduction should include the itens 2 and 3 (they should not be separeted)

Remove subtitle of item 4. Item 4.1 should be togheter the introduction. Item 4.2 - change to item 2.

Item 3. Results

It lacked about information in Table 2. It is very importante described in text about the finds in papers. Like, it was observed that in 26 studies, 25 used MTA in the comparison with others DPC agents. Only 1 study not use MTA. Follow ups ranged from 1 day to 3 months, with some studies having done only 1 folow-up  and had studies with more than 3 follow-ups. 

It was necessary to describe the main finding and the articles in a non-individualized way.

It would be important to insert in the footer of table 2 the composition of the materilas that were not described in figure 1. Or even insert in the descriptions of the types of DPC.

In figure 1, change the term solid to powder.

Discussion - Need more link with the papers in literature review and Table 2. The paragraph are very pointed, having no link with the findings in Table2.

Conclusion - It is vey superficial and only addresses the MTA. It would be important to insert the conclusions regarding other DPC agents. 

Author Response

The article is interesting and addresses an important subject in the field. However, it is necessary update references, mainly in introduction, since it is a literature review. 

Thank you! We sincerely appreciate your comments!

In introduction should include the itens 2 and 3 (they should not be separeted). Remove subtitle of item 4. Item 4.1 should be togheter the introduction. Item 4.2 - change to item 2. Item 3. Results

We took your suggestion into consideration and made these modifications.

It lacked about information in Table 2. It is very importante described in text about the finds in papers. Like, it was observed that in 26 studies, 25 used MTA in the comparison with others DPC agents. Only 1 study not use MTA. Follow ups ranged from 1 day to 3 months, with some studies having done only 1 folow-up  and had studies with more than 3 follow-ups. 

We appreciate this suggestion and we have described thouroughly the finding in the text of Results section.

It was necessary to describe the main finding and the articles in a non-individualized way.

We have added supplementary information in the discussion section which was modified and completed with connections between the results of the articles included in the review.

It would be important to insert in the footer of table 2 the composition of the materilas that were not described in figure 1. Or even insert in the descriptions of the types of DPC.

We have added some supplementary information about these materials in the footer of the tables.

In figure 1, change the term solid to powder.

We have made this modification.

Discussion - Need more link with the papers in literature review and Table 2. The paragraph are very pointed, having no link with the findings in Table2.

Indeed, we agree with you on this matter, so we’ve updated the Discussion section.

Conclusion - It is vey superficial and only addresses the MTA. It would be important to insert the conclusions regarding other DPC agents. 

We have modified the conclusion in order to make it more complex. Please find below:

From in vivo experiments on animal models, it can be clearly seen that MTA remains the most commonly used and optimum DPC agent for vital pulp therapy, either as a material in the test group or as a reference material. MTA has taken over the gold standard from CH cements, remaining today a reference standard in this group of pulp capping materials due to its properties in terms of biocompatibility, stimulation of odontoblasts, differentiation and proliferation of dental pulp cells in order to produce tertiary dentin, as shown on molecular and histological levels. As far as histological findings revealed, resin-based materials seem to be less suitable materials for pulp capping. More recently developed materials, such as bioceramic materials derived from MTA, show similar results and even surpass the ones already grounded for MTA. However, due to the diversity of animal models/teeth, type of pulp exposure, and DPC conditions, the translational success rate for some of the calcium-silicate cements might cover a wide range in human permanent/temporary teeth.